# Weak topological insulating phases
# of hard-core bosons on the honeycomb lattice

## Amrita Ghosh and Eytan Grosfeld

Department of Physics, Ben-Gurion University of the Negev, Beer-Sheva 8410501, Israel

## Abstract

We study the phases of hard-core bosons on a two-dimensional periodic honeycomb lattice in the presence of an on-site potential with alternating sign along the different $y$-layers of the lattice. Using quantum Monte Carlo simulations supported by analytical calculations, we identify a weak topological insulator, characterized by a zero Chern number but non-zero Berry phase, which is manifested at either density 1/4 or 3/4, as determined by the potential pattern. Additionally, a charge-density-wave insulator is observed at 1/2-filling, whereas the phase diagram at intermediate densities is occupied by a superfluid phase. The weak topological insulator is further shown to be robust against any amount of nearest-neighbor repulsion, as well as weak next-nearest-neighbor repulsion. The experimental realization of our model is feasible in an optical lattice setup.


# 1 Introduction

Since the discovery of topological insulators (TIs), tremendous effort has gone into the understanding of this novel phase of matter, both theoretically and experimentally [1]. The hallmark of a TI is the existence of a bulk band gap, similar to an ordinary insulator, along with protected gapless surface states. While conducting surface states can also be observed in normal band insulators, the signature that makes the TIs unique is the topological protection of the surface states by time reversal symmetry. The application prospects of TIs in spintronic devices and quantum information technology have projected them as worthy candidates for frontier research in condensed matter physics.

A three-dimensional (3D) TI is identified by four $\mathbb{Z}_2$ topological indices $(\nu_0, \boldsymbol{\nu})$, where $\nu_0$ is the strong topological index and $\boldsymbol{\nu}$ represent the weak topological indices [2–5]. A system with a non-trivial value of $\nu_0$ is known as a strong TI (STI), where gapless surface states are manifested on each two-dimensional (2D) surface of the system. On the other hand, if we try to form a 3D structure by stacking layers of 2D STIs along some particular direction, we end up with a system that exhibits gapless states on some of its 2D surfaces (depending on the stacking orientation), while the other surfaces remain gapped. This system is referred to as a weak TI (WTI) for which the strong topological index $\nu_0$ is zero, but some of the weak topological indices $\boldsymbol{\nu}$ attain non-trivial values.

In general, a $d$-dimensional WTI can be visualized as a system constructed by stacking $(d-1)$-dimensional STIs. For example, in 2D, for the case of the BDl symmetry class, according to the tenfold periodic table of topological phases [6] the strong topological index is zero, whereas a STI phase is only manifested in one-dimension (1D) with a $\mathbb{Z}$ topological invariant. If we now think of a system obtained by stacking $L_y$ 1D BDl chains with topological index $\nu$ along the $y$-direction, the resulting 2D system will behave as a WTI as long as the BDl symmetries are preserved. This system will manifest conducting edge states only along the edges localized at the two ends of the lattice in the $x$-direction. The weak topological index $\nu_x$ in such a case can be measured by averaging the strong topological index over the $L_y$ layers.

The main difference between a STI and a WTI lies in the robustness of their edge states. While for STIs symmetry-preserving disorder can never gap the edge states, this is not the situation for WTIs. Instead, the protection of the edge states in WTIs appears to require lattice-translational symmetry, so it is natural to assume that even a small amount of disorder could destroy the topological phase. However, for a 3D WTI, it was demonstrated that a conducting edge state can actually persist in the presence of disorder, as long as time-reversal symmetry and the bulk gap are preserved [7]. While the experimental verification of STIs has been performed in diverse classes of materials [8–10] since its theoretical prediction, there are only a few examples of materials exhibiting WTI phases [11–13]. Further venues are needed for the study of WTI phases [14], their stability properties [15,16], and the effects of interactions [17].

In recent years, the study of topological phases in bosonic systems has been a center of attraction in condensed matter physics. Due to the condensation property of bosons, the realization of topological phases requires interaction among the particles. This could, in fact, enhance the richness of the various topological phases observed in a bosonic system. Moreover, recent advancements in optical lattice experiments have created a promising platform, where different phases of interacting and non-interacting bosonic systems can be realized in a controlled manner. These developments highlight the need for an extensive theoretical study of topological phases of bosons in the presence of interactions. In particular, a study of interacting bosonic analogues of WTIs can help identify natural, minimal models that nucleate such phases, explore the interplay of the various competing orders that arise in such systems, and determine the effect of interactions on the emerging phase diagram.

In this paper we study the infinite on-site repulsion limit of bosons [hard-core bosons

(HCBs)] on the 2D honeycomb lattice in the presence of on-site potential and longer-range interactions. We demonstrate that weak topological phases arise quite naturally when the HCBs are simply subjected to an on-site potential with alternating signs along the different $y$-layers. Using quantum Monte Carlo (QMC) technique supported by analytical calculations, we find that the phase diagram of the model exhibits three insulating phases at densities 1/4, 1/2 and 3/4, separated from each other by a superfluid region. Depending on the choice of the on-site potential form, either the insulator at 1/4 or 3/4 filling is found to be a WTI, which manifests a nontrivial Berry phase and the existence of edge states along the $x$-edges of the lattice. These WTI phases away from half-filling are a prime example of a mirror-protected WTI. We introduce a formula for the Berry phase that relies on the permanent (rather than a determinant) for the HCBs, and uncover a robust 1D superfluidity along the topological edge states. Finally, we demonstrate a remarkable stability of the topological phase against any amount of nearest-neighbor (NN) repulsion, as well as weak next-nearest-neighbor (NNN) repulsion among the HCBs. Through these developments we introduce a framework that could precipitate the study of additional bosonic TIs, see, e.g., [18].

The paper is organized as follows. In Section 2 we present the model, the numerical techniques and the relevant order parameters. In Section 3 we present the phase diagram of the model and analyze the different phases. The edge states of the insulating phases of the model are analyzed using QMC methods in Section 4. In Section 5 we calculate the topological invariants for the insulating phases. Next, in Section 6 the effect of NN and NNN repulsion on the WTI is presented. Lastly, in Section 7 we conclude. In Appendices A and B we analyze the band structure of the model and discuss the protection of the edge states.

## 2  Model and Formulation

We consider HCBs in a 2D periodic honeycomb lattice, as depicted in Fig. 1, governed by the Hamiltonian

$$\hat{H} = -t \sum_{\langle i,j \rangle} \left( \hat{d}_i^\dagger \hat{d}_j + \text{h.c.} \right) + \sum_i W_i \hat{n}_i - \sum_i \mu \hat{n}_i. \tag{1}$$

Here $\hat{d}_i^\dagger$ ($\hat{d}_i$) creates (annihilates) a HCB at site $i$, $\langle i,j \rangle$ represent NN pairs of sites, $t$ is the amplitude of NN hopping, $W_i$ is the on-site potential at site $i$ and $\mu$ denotes the chemical potential. We take the NN hopping as the unit of energy and set $t = 1$ for our numerical calculations. In our study $W_i$ forms a periodic potential along the $y$-direction with a period of two lattice sites, i.e., we take $W_i = W_0 (-W_0)$ for layers (along the $y$-direction) labeled by odd (even) values of $\ell$. We shall assume that the lattice constant is $a = 1$ throughout.

To study the various phases of the Hamiltonian in Eq. (1), we use the Stochastic-Series-Expansion (SSE) technique [19, 20], a quantum Monte Carlo method, employing directed loop updates [21, 22]. To capture the ground state-properties of a $L \times L$ honeycomb lattice using SSE, all simulations have been done at low enough temperatures such that the inverse temperature $\beta \sim L$ [23].

To construct the phase diagram using SSE we use four order parameters: average density $\rho$, superfluid density $\rho_s$, structure factor $S(\boldsymbol{Q})$ and dimer structure factor $S_D(\boldsymbol{Q})$.

The average density of a system containing $N_s$ sites is $\hat{\rho} = \sum_i \hat{n}_i / N_s$, where $\hat{n}_i = \hat{d}_i^\dagger \hat{d}_i$ gives the number of HCBs (either 0 or 1) at site $i$. To calculate the superfluid density using SSE, we employ the following expression in terms of the winding numbers $\Omega_x$ and $\Omega_y$ along $x$ and $y$-directions [20],

$$\rho_s = \frac{1}{2\beta} \left\langle \Omega_x^2 + \Omega_y^2 \right\rangle \equiv \rho_s^x + \rho_s^y, \tag{2}$$

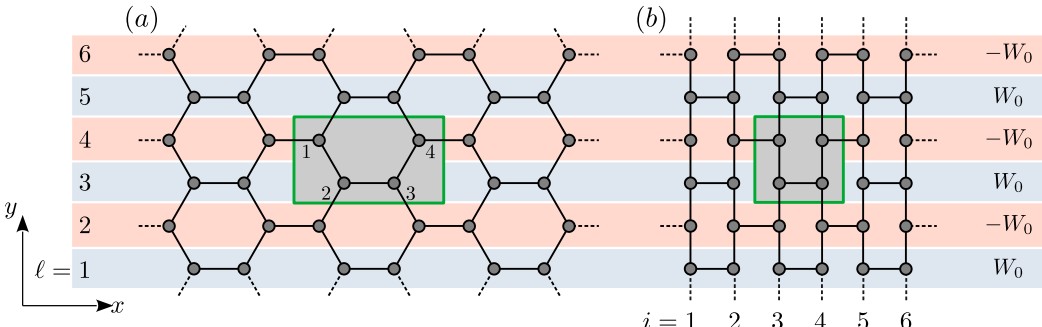

Figure 1: Schematic of the 2D periodic lattices considered in the main text: (a), the honeycomb lattice with alternating values of on-site potential along the $y$-direction; and, (b), the transformed version obtained by straightening the bonds along the $y$-axis. The $y$-layers are indexed by $\ell = 1, 2, \ldots$ and are highlighted by blue (red) backgrounds indicating the presence of an on-site potential $W_0 (-W_0)$ along the layer. The dashed black lines represent the bonds connecting the lattice across the boundaries. Each unit cell, delineated by a green rectangle, contains four sites labeled by the numbers 1 to 4. The stripes along the $y$-direction are indexed by $i = 1, 2, \ldots$ in panel (b).

where $\langle \cdots \rangle$ represents ensemble average. For example, the winding number $\Omega_x$ can be calculated by counting the total number of operators $N_x^+ (N_x^-)$ transporting particles in the positive (negative) $x$-direction, according to the formula $\Omega_x = \frac{1}{L_x}(N_x^+ - N_x^-)$, where $L_x$ is the length of the lattice along the $x$-direction.

Next, the structure factor per site is expressed as,

$$S(\boldsymbol{Q}) = \frac{1}{N_s^2} \sum_{i,j} e^{i\boldsymbol{Q} \cdot (\boldsymbol{r}_i - \boldsymbol{r}_j)} \langle \hat{n}_i \hat{n}_j \rangle, \tag{3}$$

where $\boldsymbol{r}_i = (x_i, y_i)$ is the position of site $i$. To calculate the structure factor for particles in a $L \times L$ honeycomb lattice, we can always use a transformation on the lattice to straighten the bonds along the $y$-direction, such that the resulting lattice looks like the one depicted in Fig. 1 b. With the use of the position vectors $\boldsymbol{r}$ of this new transformed lattice, the allowed values of the wavevector $\boldsymbol{Q}$ coincide with those of an $L \times L$ square lattice, i.e., $\boldsymbol{Q} = (2\pi p/L, 2\pi q/L)$, where $p = 0, 1, \cdots, L-1$ and $q = 0, 1, \cdots, L-1$. To detect the presence of diagonal long-range orders in the system, we have calculated $S(\boldsymbol{Q})$ for all possible values of $\boldsymbol{Q}$ and identify the ones at which the structure factor displays peaks.

Lastly, we define the dimer structure factor as

$$S_D(\boldsymbol{Q}) = \frac{1}{N_b^2} \sum_{\alpha, \beta} e^{i\boldsymbol{Q} \cdot (\boldsymbol{R}_\alpha - \boldsymbol{R}_\beta)} \langle \hat{D}_\alpha \hat{D}_\beta \rangle, \tag{4}$$

where $\hat{D}_\alpha = \hat{d}_{\alpha_L}^\dagger \hat{d}_{\alpha_R} + \hat{d}_{\alpha_R}^\dagger \hat{d}_{\alpha_L}$ is the dimer operator defined on the $\alpha$-th NN bond aligned along $x$-axis with $\alpha_L$, $\alpha_R$ being the two lattice sites attached to this bond. In Eq. (4), the summation runs over $N_b$ NN bonds oriented along $x$-axis and the vectors $\boldsymbol{R}$ represent the position coordinate corresponding to the midpoints of these bonds in the transformed lattice in Fig. 1 b. The dimer operator is chosen in a way such that it will give a nonzero expectation value only when a dimer is formed, i.e., when the constituent particle hops back and forth along the NN bond.

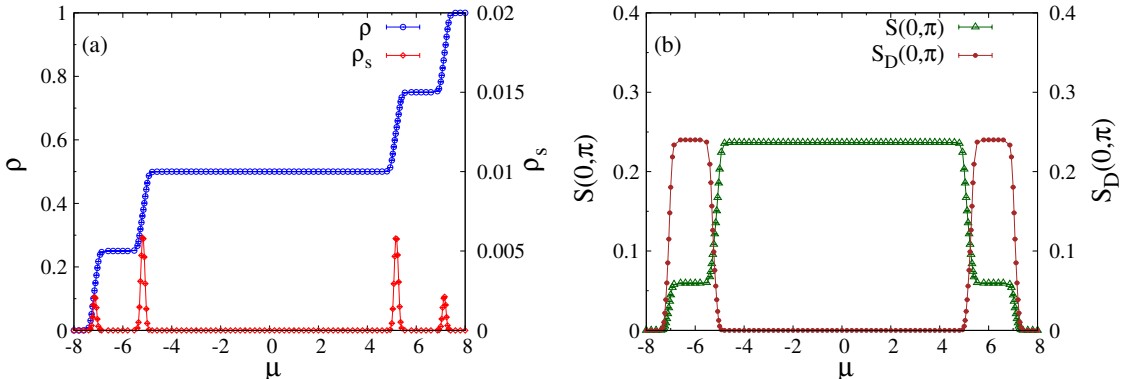

Figure 2: Plots of the four order parameters for $W_0 = 6.0$ as a function of the chemical potential $\mu$: (a), density $\rho$ and superfluid density $\rho_s$; and, (b), structure factor $S(0, \pi)$ and dimer structure factor $S_D(0, \pi)$. Here $t = 1.0$ and the calcualtions are performed on a $20 \times 20$ periodic honeycomb lattice.

## 3 Phase diagram

To construct the phase diagram we study the Hamiltonian in Eq. (1) at various values of $W_0$ by varying the chemical potential $\mu$. Fig. 2 depicts the variations of HCB density $\rho$, superfluid density $\rho_s$, structure factor $S(0, \pi)$ and dimer structure factor $S_D(0, \pi)$ as a function of $\mu$, where the value of $W_0$ is fixed at 6.0. The three plateaus in the $\rho - \mu$ curve (apart from the trivial ones at $\rho = 0$ and $\rho = 1$) clearly indicate the presence of three incompressible insulators at densities 1/4, 1/2 and 3/4. At these plateaus the superfluid density becomes zero, whereas in the intermediate regions it attains some non-zero value, thus separating the three insulating regions by a superfluid phase. To understand the nature of the insulators we have calculated the structure factor $S(Q)$ and dimer structure factor $S_D(Q)$ for all possible values of $Q$. We find that $S(Q)$ peaks only at wavevector $Q = (0, \pi)$, whereas $S_D(Q)$ displays peaks for both $Q = (0, \pi)$ and $Q = (\pi, 0)$ with the same peak value (in Fig. 2 b only $S_D(0, \pi)$ is displayed).

We note that the results in Fig. 2 are independent of the sign of $W_0$, i.e., whether we choose $W_i$ to be positive (negative) for odd (even) values of $\ell$ in Fig. 1 or the reverse scenario, Fig. 2 remains unaltered. In the following, we assume $W_0 > 0$ to analyze our results, but a similar analysis can be extended to the reverse scenario as well. In Section 4 we will see that the sign of $W_0$ nevertheless plays a role in the characterization of the different phases.

Since the on-site potential for the layers labeled by even values of $\ell$ is $-W_0$, upto half-filling the particles will prefer to occupy these layers only, keeping the odd $\ell$ layers completely empty. Due to the presence of NN hopping, at 1/4-filling, it is energetically favorable for the system to fill the upper two sites of each unit cell (i.e., site 1 and 4) by one particle only, so that this particle can hop back and forth between sites 1 and 4 of two adjacent unit cells to further lower the energy of the system. As a result of this hopping process dimers are formed between two sites belonging to the upper half of two different unit cells. Due to the formation of these dimers there is no net flow of HCBs in the $x$ or $y$-directions, which makes the phase insulating in nature. The structure of this dimer insulator at 1/4-filling is depicted in Fig. 3 a. We note that at each even $\ell$ level we have one dimer which involves two boundary sites when the open boundary condition is applied along $x$-direction with zigzag edges.

Now, let us analyze the structure factor, Eq. (3), for $Q = (0, \pi)$,

$$S(0, \pi) = \frac{1}{N_s^2} \sum_{i,j} e^{i\pi(y_i - y_j)} \langle \hat{n}_i \hat{n}_j \rangle. \tag{5}$$

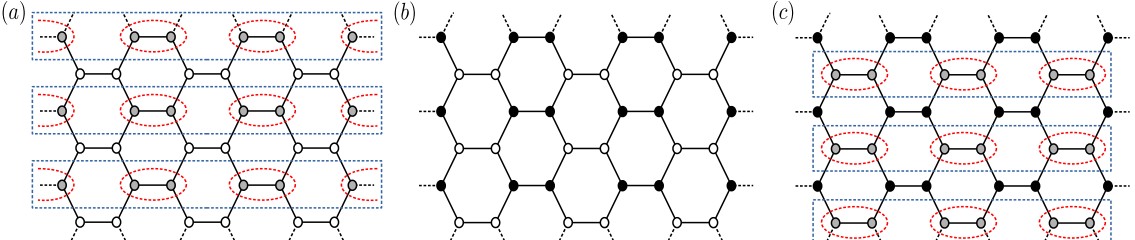

Figure 3: Spatial structures of the insulating phases: (a) The dimer insulator at density $\rho = 1/4$; (b) Charge-density-wave insulator at half-filling; and, (c) The dimer insulator at filling fraction 3/4. The red dashed lines depict the formation of dimers. The white, grey and black circles signify lattice sites with density 0.0, 0.5 and 1.0 respectively. The blue dashed rectangles represent the underlying 1D SSH-like chains, at their, (a), topological and, (c), non-topological phases.

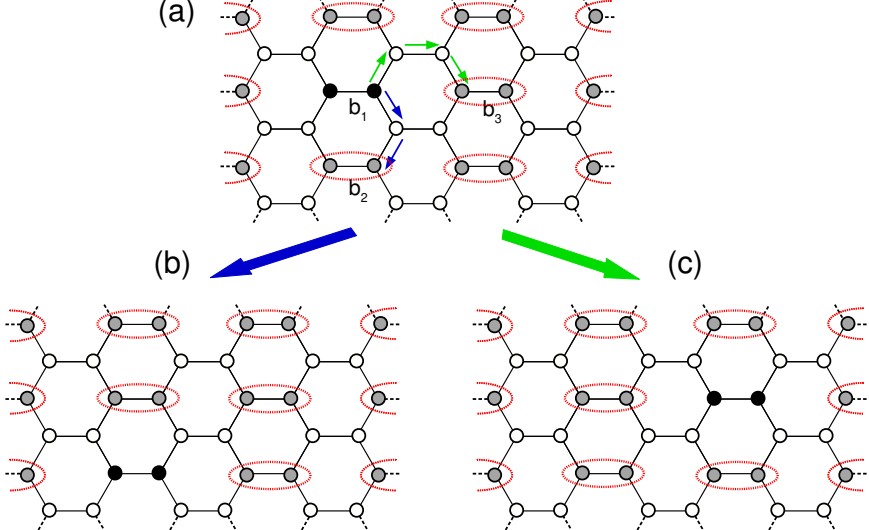

Figure 4: Pictorial description of hopping processes in the superfluid region in-between the dimer insulator at $\rho = 1/4$ and CDW structure at $\rho = 1/2$.

Although the summation in Eq. (5) is over all possible pairs of sites in the lattice, only those pairs for which both sites are occupied will have a non-zero contribution. Since for the dimer insulator at $\rho = 1/4$ (see Fig. 3 a), all particles reside on the even layers only, for all contributing pairs $y_i - y_j$ is even, so Eq. (5) reduces to

$$S(0, \pi) = \frac{1}{N_s^2} \sum_{i,j} \langle \hat{n}_i \hat{n}_j \rangle. \tag{6}$$

At 1/4-filling there are $N_s/4$ particles in the system and each of them participates in $N_s/4$ pairs in the summation (including the case where $i = j$) with a +1 contribution to the structure factor. Therefore, for the dimer insulator at $\rho = 1/4$, $S(0, \pi)$ attains the value

$$S(0, \pi) = \frac{1}{N_s^2} \left( \frac{N_s}{4} \right)^2 = 0.0625. \tag{7}$$

This result matches well with the result in Fig. 2 b. It is clear from the discussion above that at 1/4-filling, as long as the particles are constrained to reside on alternate layers, the value of

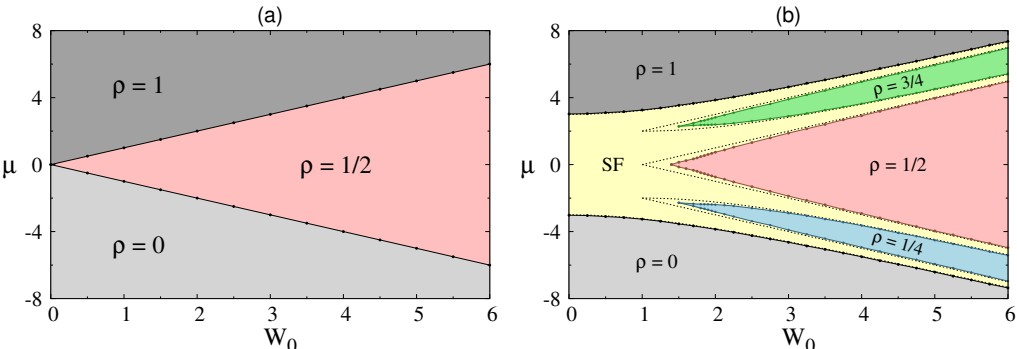

Figure 5: Phase diagram of HCBs on the honeycomb lattice as function of the chemical potential ($\mu$) and the alternating on-site potential strength along the $y$-direction ($W_0$): (a) in the atomic limit; and, (b) in the presence of a finite hopping $t = 1.0$. The $\rho = 0$ (light grey), $\rho = 1/2$ (pink) and $\rho = 1$ (dark grey) regions denote the empty phase, the charge-density-wave insulator at half-filling and the Mott insulator at filling-fraction 1, respectively. The blue (green) regions indicate the dimer insulator at 1/4 (3/4)-filling. The yellow region depicts the superfluid phase. The solid lines with points indicate phase boundaries obtained using SSE (for $20 \times 20$ lattice with $\beta = 120$), whereas the dashed lines indicate the calculated band edges.

$S(0, \pi)$ will be 0.0625. This value is independent, e.g., of whether the particles form a dimer insulator or arrange themselves in a charge-density-wave (CDW) pattern.

To manifest the formation of dimer insulator at density $\rho = 1/4$, we next calculate the dimer structure factor $S_D(\mathbf{Q})$ as prescribed in Eq. (4). Since in our system the dimers are formed along the NN bonds oriented along $x$-direction of the lattice, we have defined the dimer structure factor such that it will detect dimers along these bonds only. Now, if we think about the dimer-insulator structure corresponding to $\rho = 1/4$ (as depicted in Fig. 3 a) for the transformed lattice in Fig. 1 b, it is easy to see that for any two dimers with midpoints $\mathbf{R}_\alpha = (X_\alpha, Y_\alpha)$ and $\mathbf{R}_\beta = (X_\beta, Y_\beta)$, both $(X_\alpha - X_\beta)$ and $(Y_\alpha - Y_\beta)$ are even. As a result, the dimer structure factors for $\mathbf{Q} = (0, \pi)$ and $\mathbf{Q} = (\pi, 0)$ reduce to the exact same expression,

$$S_D(0, \pi) = S_D(\pi, 0) = \frac{1}{N_b^2} \sum_{\alpha, \beta} \langle \hat{D}_\alpha \hat{D}_\beta \rangle. \tag{8}$$

In the dimer insulator phase, with $N_b$ being the total number of NN bonds along $x$-direction, there are $N_b/2$ dimers in the system. Each of these dimers will participate in $N_b/2$ pairs (of dimers) in the summation, with a +1 contribution towards the dimer structure factor. Therefore, the dimer structure factor reduces to,

$$S_D(0, \pi) = S_D(\pi, 0) = \frac{1}{N_b^2} \left( \frac{N_b}{2} \right)^2 = 0.25, \tag{9}$$

which is indeed attained in Fig. 2 b at filling 1/4.

Next, at 1/2-filling, the layers labeled by even $\ell$ values are completely filled, such that the upper two sites of each unit cell are occupied by two HCBs. Therefore, at this density the dimers of 1/4-filling disappear completely and we have a CDW, similar to the one depicted in Fig. 3 b, which is insulating in nature. This can further be verified from Fig. 2 b, where we can see that at $\rho = 1/2$ the dimer structure factor $[S_D(0, \pi)]$ vanishes and the structure factor $[S(0, \pi)]$ shows a peak with value 0.25. The half-filled system contains $N_s/2$ particles in total and each of them participates in $N_s/2$ pairs of sites, which has a non-zero contribution towards

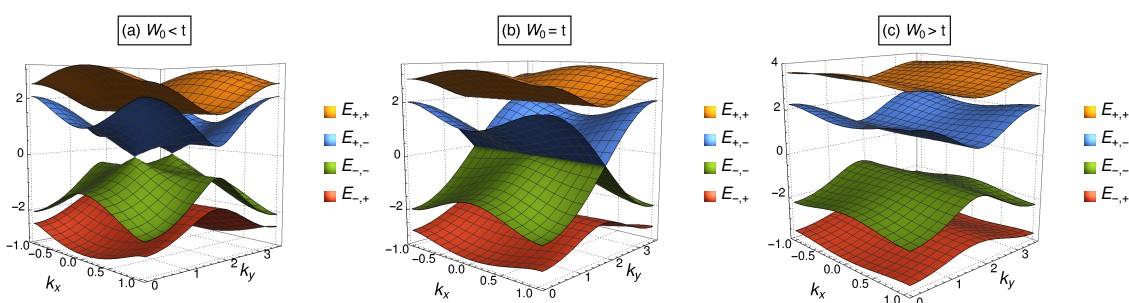

Figure 6: The four branches of energy spectrum corresponding to (a) $W_0 < t$, (b) $W_0 = t$, and (c) $W_0 > t$.

the structure factor. So, Eq. (6) in this case simply becomes

$$S(0, \pi) = \frac{1}{N_s^2} \left( \frac{N_s}{2} \right)^2 = 0.25, \tag{10}$$

which is the maximum value attained by $S(0, \pi)$.

Finally, the structure of the insulator at 3/4-filling is depicted in Fig. 3 c, where the even $\ell$ levels are completely filled and the odd ones are half-filled. In terms of unit cells this means that the upper two sites (sites 1 and 4) of each cell are fully occupied and the lower two sites (sites 2 and 3) share a single HCB. Again by virtue of NN hopping, the particles at odd $\ell$-levels can further lower their energy by hopping back and forth between sites 2 and 3 of each unit cell. As a result, dimers are formed in the lower half of each unit cell. The main difference between the dimers formed at $\rho = 1/4$ and $\rho = 3/4$ is that, the dimers at 1/4-filling are formed between two sites belonging to two different unit cells, whereas the sites involved in 3/4-filling are residents of the same unit cell. Since the number of dimers formed in this case coincides with the one for $\rho = 1/4$, the dimer structure factor attains the same peak value 0.25 in this situation as well. The value of the corresponding structure factor can be extracted by realizing that out of the $3N_s/4$ particles in the system, $N_s/4$ reside on the odd $\ell$ layers, whereas $N_s/2$ particles are located at even $\ell$ layers. So, in total there are $2(N_s/4)(N_s/2)$ pairs for which the separation between the particles along the $y$-axis (i.e., $y_i - y_j$ in Eq. (5)) is odd. Clearly each of these pairs will contribute $-1$ to the structure factor (as $e^{i\pi(y_i - y_j)} = -1$ for these cases). On the other hand, for $(N_s/4)^2 + (N_s/2)^2$ number of pairs, $y_i - y_j$ is an even multiple of the lattice constant, which gives rise to a positive contribution to the structure factor. Therefore, the structure factor for this insulator attains the value,

$$\begin{aligned} S(0, \pi) &= \frac{1}{N_s^2} \left[ \left( \frac{N_s}{4} \right)^2 + \left( \frac{N_s}{2} \right)^2 - 2 \left( \frac{N_s}{4} \right) \left( \frac{N_s}{2} \right) \right] \\ &= \frac{1}{N_s^2} \left( \frac{N_s}{2} - \frac{N_s}{4} \right)^2 = 0.0625, \end{aligned} \tag{11}$$

which coincides with the one for 1/4-filling, Eq. (7).

Next we turn to discuss the superfluid phase. In the intermediate regions between the insulating phases, where the superfluid density is finite, both the structure factor $S(0, \pi)$ and dimer structure factor $S_D(0, \pi)$ admit nonzero values, see Fig. 2. Interestingly, in these intermediate regions, we observed anisotropy in the superfluid density where $\rho_s^y$, the superfluid density along $y$-direction of the lattice, is much larger than the one along the $x$-direction, $\rho_s^x$. So while the superfluid retains some additional structure from the two neighboring insulators

in the transition region, it can still superflow in both directions, as we now argue. The mechanism is described in Fig. 4, where we take for example the transition between the insulators at fillings 1/4 and 1/2. Consider a situation where we add one HCB to the dimer insulator at $\rho = 1/4$. At this range of fillings the particles will naturally prefer to occupy the even layers, having the lower on-site potential, so the extra particle chooses to reside on one of the sites attached to bond $b_1$ in Fig. 4 a, effectively generating a doubly occupied bond. Now, this extra particle can hop through the lattice giving rise to superfluidity in two different ways. Firstly, the extra particle can follow a two-step hopping process similar to the one depicted by blue arrows in Fig. 4 a. By virtue of this process, effectively the doubly occupied bond $b_1$ has hopped to bond $b_2$ (Fig. 4 b) giving rise to superfluidity along $y$-direction. Since the difference of energies of the particle at the initial (or final) and intermediate step of this process is $2W_0$, the energy gained by the particle during this process is $\sim t^2/(2W_0)$. Secondly, the particle can also follow a three-step hopping process depicted by the green arrows in Fig. 4 a. This process results in a configuration as shown in Fig. 4 c, where the doubly occupied bond $b_1$ has effectively hopped along $x$-direction of the lattice to the bond $b_3$, contributing to a non-zero superfluid density $\rho_s^x$. On account of the fact that both the intermediate sites involved in this hopping process have energies higher than the initial or final sites, by an amount of $2W_0$, one can see that the energy gain in this process is $\sim t^3/(4W_0^2)$. Therefore, in comparison the doubly occupied bond can always gain more energy by hopping in the $y$-direction of the lattice than in the $x$-direction. Consequently, anisotropy is developed in the superfluid density with $\rho_s^y > \rho_s^x$. Similar arguments hold for the superfluid regions between any two insulators.

The complete phase diagram of the model in the $(\mu, W_0)$ plane is depicted in Fig. 5 in the atomic limit, i.e., when the hopping $t$ is turned off (Fig. 5 a) and for $t = 1.0$ (Fig. 5 b). The phase boundaries are obtained from QMC (solid lines with points), performed for a $20 \times 20$ periodic honeycomb lattice with inverse temperature $\beta = 120$.

In the presence of finite NN hopping $t = 1.0$ (Fig. 5 b), for $W_0 = 0$, the superfluid phase fills the range between the Mott lobes at densities $\rho = 0$ and $\rho = 1$. Beyond some critical value of $W_0$, additional insulating lobes start to appear at densities 1/4, 1/2 and 3/4 separated by superfluid regions. The insulator at half-filling is a CDW, whereas the other two are dimer insulators. One can see that the phase boundaries obtained from QMC are more or less consistent with the calculated band edges from Appendix A (dashed lines in Fig. 5 b), except in the neighborhood of $W_0 = 1$, where they slightly deviate. Since the analytical results demonstrate the phase boundaries for a lattice in the thermodynamic limit, we expect the deviations to be smaller for larger system sizes. Within the error bars of our QMC calculations the critical value of $W_0$ beyond which the insulating phases appear, is 1.4 for the CDW at half-filling, whereas for the dimer insulators it appears to be 1.5. However, the calculated band edges predict that the tips of all three insulating lobes lie at $W_0 = 1$.

Next, in the atomic limit (Fig. 5 a), we see that only the CDW insulator at half-filling survives and both the dimer insulators vanish completely. Indeed, the dimers in the dimer insulators are formed by virtue of the NN hopping in the presence of a finite $W_0$. On the other hand, in the atomic limit the CDW insulator appears as soon as we have a non-zero $W_0$. In fact, the presence of NN hopping can destroy this structure by transforming it into a superfluid, as is indeed observed in Fig. 5 b. Beyond some critical value of $W_0$ the CDW phase sets in because at this stage $W_0$ dominates over hopping $t$ and it becomes energetically favorable for the particles to be frozen in this structure instead of moving around in a superfluid phase. The boundaries of the CDW phase in Fig. 5 a can be determined by considering the change in the total energy of the system when we introduce an additional HCB in the half-filled system manifesting the CDW phase. At half-filling all the particles occupy the sites with on-site potential $-W_0$. As a result, in this situation the total energy of the system is simply $E[N_s/2] = -\mu N_s/2 - W_0 N_s/2$. Now, if we try to add another HCB to the system,

this additional particle will have to occupy a site with on-site potential $W_0$. So, in this scenario the total energy of the system will be $E[N_s/2 + 1] = -\mu(N_s/2 + 1) - W_0 N_s/2 + W_0$. Thus, the change in the total energy of the system to add an additional HCB is $\Delta E = -\mu + W_0$. As long as $\Delta E > 0$ the phase remains stable against the addition of an extra particle; therefore, the upper phase boundary of the CDW is given by the line $\mu = W_0$. Similarly the lower phase boundary can be determined by following the same procedure for the case when we reduce one particle from the half-filled system, for which the boundary will be given by the line $\mu = -W_0$.

The calculated band edges (see Appendix A) appear as dashed lines in Fig. 5 b. The nature of the different phases is further elucidated by the calculated spectrum, presented in Fig. 6. For $W_0 < t$ a Dirac semimetal is observed at $\rho = 1/2$, while for $W_0 = t$ it is replaced by a nodal line semimetal. Finally, for $W_0 > t$, the phases at $\rho = 1/4$ and $\rho = 3/4$ develop a full gap, and the dimer insulators are formed. Since the dimer insulators have no corresponding atomic limits, they appear to be more interesting to investigate. In the next section, we shed some light on the nature of these insulators by exploring their edge structure.

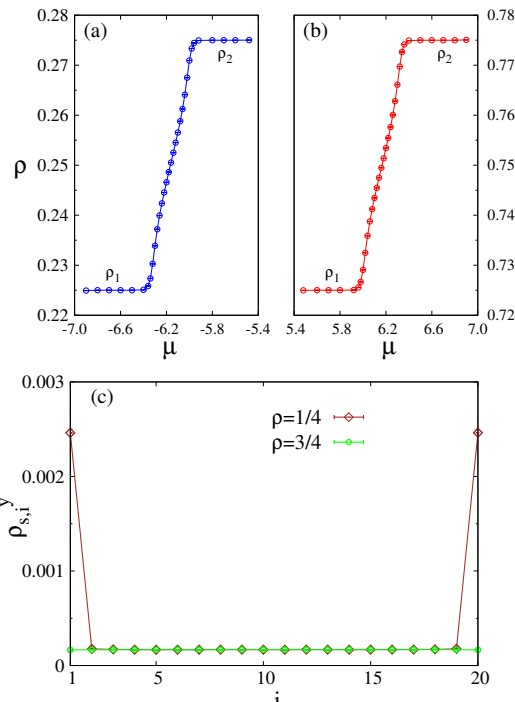

Figure 7: Splitting of the plateau at: (a), $\rho = 1/4$ with $W_0 = 6.0$ ; and, (b), $\rho = 3/4$ at $W_0 = -6.0$, when open boundary conditions are applied along the $x$-direction on a $20 \times 20$ honeycomb lattice. (c) The superfluid density $\rho_{s,i}^y$ for different stripes ($i$) of the same $20 \times 20$ open lattice measured for $\mu = -6.16$ ($\rho = 1/4$) and $\mu = 6.16$ ($\rho = 3/4$) with $W_0 = 6.0$. Here we take $t = 1.0$ and $\beta = 120$.

## 4 Edge States

To further explore the nature of the dimer insulators, we measure the shift in the average density $\rho$ when we switch to open boundary conditions along the $x$-direction, i.e., by turning

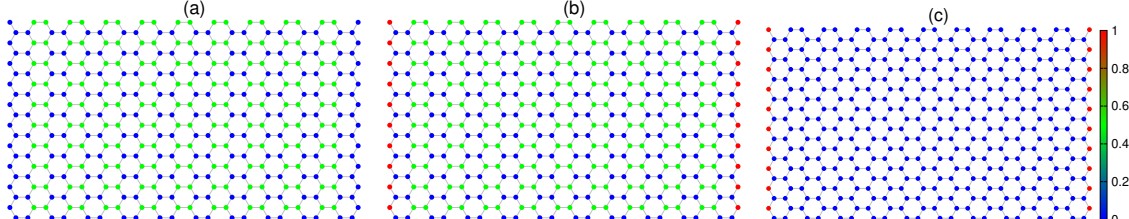

Figure 8: Local density profile of a $20 \times 20$ honeycomb lattice under open boundary condition along $x$-direction with $W_0 = 6$, $t = 1$ and $\beta = 120$, corresponding to: (a) $\mu = -6.5$ in the lower plateau, (b) $\mu = -5.8$ in the upper plateau, and (c) the difference between densities of (b) and (a).

off the horizontal dashed bonds in Fig. 1 b [24, 25]. We observe that as a function of the chemical potential $\mu$ the average density $\rho$ remains unaltered except for $\rho = 1/4$, which splits into two plateaus corresponding to densities $\rho_1 = 0.225$ and $\rho_2 = 0.275$, see Fig. 7 a. Further investigation reveals that the values of $\rho_1$ and $\rho_2$ depend on the size of the system: for an $8 \times 8$ system $\rho_1 = 0.1875$ and $\rho_2 = 0.3125$, whereas for a $10 \times 10$ system $\rho_1 = 0.2$ and $\rho_2 = 0.3$. Generally speaking, we find that for a honeycomb lattice with $N_e$ edge sites (depicted by red circles in Fig. 8 c) and $N_s$ total number of sites, the plateau at $\rho = 1/4$ splits into $\rho_1 = \rho - N_e/(2N_s)$ and $\rho_2 = \rho + N_e/(2N_s)$.

We now argue that the splitting of the plateau under open boundary conditions corresponds to the number of in gap edge states. At 1/4-filling the dimer structure can be thought of as a series of 1D Su-Schrieffer-Heeger (SSH)-like chains (depicted by blue dashed rectangle in Fig. 3 a), where dimers of strength $\sim t$ are formed. The dimers in each chain are weakly coupled to each other via third-order hopping through the intermediate sites with higher on-site potential, $t_x \sim t^3/(2W_0^2)$. Now, each of these SSH-like chains will give rise to a pair of degenerate in-gap edge-states under open boundary condition, localized at the two ends of the chain. As is clear from Fig. 3 a, for a $L_x \times L_y$ lattice, there are $L_y/2$ different SSH chains weakly connected to each other via second order hopping, $t_y \sim t^2/(2W_0)$. Due to this inter-chain interaction, the degeneracy of the edge-states will be lifted and the resulting $L_y$ in-gap states will now form bands on the two edges, each with a bandwidth $\sim 4t_y$. The two plateaus therefore correspond to the situation when: (1), none of the edge sites are occupied; and, (2), all of the edge sites are completely occupied, beyond some critical chemical potential determined by the edge bandwidth. In our notations this reduces to $\rho_1 = \rho - 1/(2L_x)$ and $\rho_2 = \rho + 1/(2L_x)$.

To help visualize this, in Fig. 8 a, we plot the local density profile of a $20 \times 20$ honeycomb lattice under open boundary condition along the $x$-direction, by choosing a value for chemical potential, $\mu = -6.5$, in the lower plateau $\rho_1$. We can clearly see that the lower plateau corresponds to a situation where bulk sites of the even $\ell$-layers have density 0.5 each, while the edge sites have density close to zero. Contrarily, Fig. 8 b depicts the density profile corresponding to $\mu = -5.8$, a point in the upper plateau $\rho_2$. The density of the edge sites now becomes close to 1, while the density of the bulk sites remains 0.5 as before. So, at the $\rho_1$ ($\rho_2$) plateau, each of the edge sites have 1/2 HCB less (more) compared to the sites in the bulk. The difference of these two local density profiles is shown in Fig. 8 c, which clearly demonstrates that the transition between the two plateaus, in Fig. 7 a, corresponds to the occupation of the in-gap edge states. All this indicates towards the possible topological nature of the dimer insulator at 1/4 filling-fraction.

Next, we study the effect of the reversal of the sign of $W_0$. In this case, at 1/4-filling, the dimers are formed at odd $\ell$-layers and hence no dimers are split by the opening of the boundary conditions [see Fig. 3 c with black (filled) sites replaced by white (empty) sites]. On the other hand, at $\rho = 3/4$, all the sites residing on the odd $\ell$-layers are completely occupied and the dimers are formed in the even $\ell$-layers [see Fig. 3 a with white (empty) sites replaced by black (filled) sites]. Therefore, in this case, it is the dimer insulator at 3/4 filling which involves the formation of edge states. As a result, under open boundary condition along $x$-direction, the plateau corresponding to $\rho = 3/4$ splits into two plateaus (as shown in Fig. 7 b) corresponding to densities $\rho_1 = 0.725$ and $\rho_2 = 0.775$, while the other parts of the $\rho - \mu$ curve remain unchanged.

In Fig. 7 c we study the superfluid density, $\rho_{s,i}^y$, for the different stripes ($i$) along the $y$-direction (see Fig. 1 b) of a $20 \times 20$ lattice with open boundary conditions along $x$-direction with $W_0 = 6.0$. The superfluid density is measured for two values of the chemical potential; $\mu = -6.16$ corresponding to the density $\rho = 1/4$ in Fig. 7 a, which ensures a partial occupation of the edge states, and $\mu = 6.16$ corresponding to the middle of the unsplit density plateau

at $\rho = 3/4$. For $\mu = -6.16$ the superfluid density acquires a non-zero value at the two ends of the lattice, while within the bulk of the lattice it acquires a vanishing value. On the other hand for $\mu = 6.16$ the superfluid density remains vanishingly small for all the stripes. In the thermodynamic limit while the superfluid density tends to zero for the bulk stripes at $\rho = 1/4$, it remains finite at the edges. This captures the conducting nature of the edge states.

Finally, we note that by opening the boundaries of the lattice along the $y$-direction, we end up with a honeycomb lattice with armchair edges and no dimers are split by this change. Therefore, in this case the $\rho - \mu$ curve remains unaffected by the open boundary condition for both positive and negative $W_0$. Thus, edge states are manifested only along the zigzag edges in the $x$-direction.

## 5 The topological invariant

In order to confirm the topological nature of the dimer insulators from a bulk perspective, one must calculate the topological invariant for the system under periodic boundary conditions. We now turn in this direction. For a $L_x \times L_y$ honeycomb lattice, the lattice points on the discrete Brillouin zone $(k_x, k_y)$ can be expressed as,

$$k_x = -\frac{\pi}{3} + \frac{2\pi}{3L_x} p, \qquad p = 1, 2, \cdots, L_x, \qquad (12)$$

$$k_y = \frac{2\pi}{\sqrt{3}L_y} q, \qquad q = 1, 2, \cdots, L_y. \qquad (13)$$

Let $\psi_{\pm,\pm}$ denote the normalized column eigenvectors corresponding to the energy bands $E_{\pm,\pm}$ of Eq. (21) (see Appendix A). Then the ground state multiplets $\Psi^{(\rho)}$ corresponding to filling-fractions 1/4, 1/2 and 3/4 are given by the matrices $\Psi^{(1/4)} = \{\psi_{-,+}\}$, $\Psi^{(1/2)} = \{\psi_{-,+}, \psi_{-,-}\}$ and $\Psi^{(3/4)} = \{\psi_{-,+}, \psi_{-,-}, \psi_{+,-}\}$. One can then define the U(1) link variables along the two directions as,

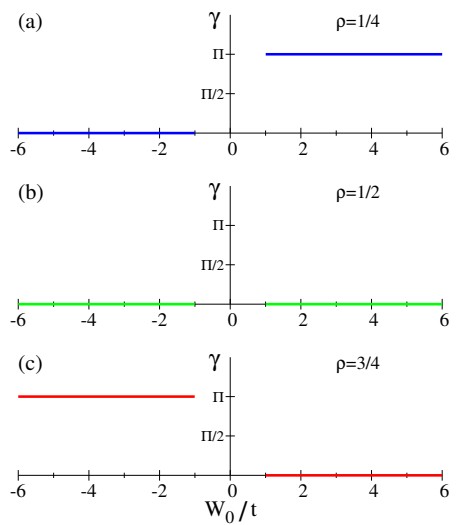

Figure 9: The calculated Berry phase $\gamma$ along the $x$-direction of the Brillouin zone as a function of $W_0/t$ for the three gapped phases at densities (a) 1/4, (b) 1/2 and (c) 3/4.

$$U_x(p,q) = \frac{\det_{\pm}\left(\Psi_{p,q}^{\dagger}\Psi_{p+1,q}\right)}{\left|\det_{\pm}\left(\Psi_{p,q}^{\dagger}\Psi_{p+1,q}\right)\right|}, \qquad (14)$$

and

$$U_y(p,q) = \frac{\det_{\pm}\left(\Psi_{p,q}^{\dagger}\Psi_{p,q+1}\right)}{\left|\det_{\pm}\left(\Psi_{p,q}^{\dagger}\Psi_{p,q+1}\right)\right|}, \qquad (15)$$

where $\det_+$ ($\det_-$) corresponds to the permanent (determinant) of the matrix. It is important to note that in the definition of the link variable, the permanent is applicable when the particles under consideration are HCBs, whereas for fermions $U$ involves determinant. Finally, the Chern number can be calculated as,

$$C = \frac{1}{2\pi i} \sum_{p,q} F_{xy}(p,q), \qquad (16)$$

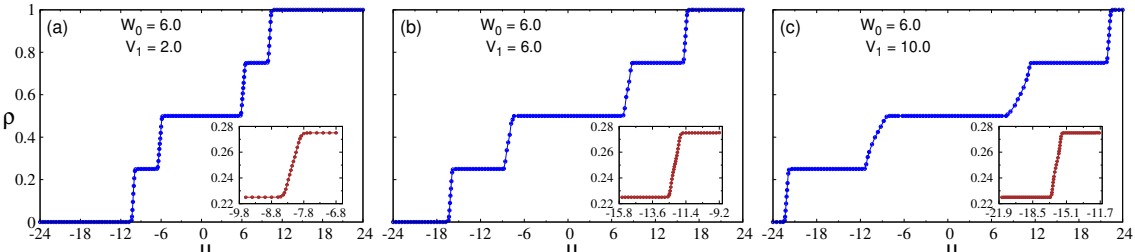

Figure 10: Variations of density versus chemical potential for three different values of NN repulsion $V_1$, with the on-site potential strength $W_0 = 6.0$. The blue curves show the $\rho - \mu$ variations for a $20 \times 20$ honeycomb lattice with periodic boundary conditions. The brown curves in the insets display the splitting of the $\rho = 1/4$ insulator under open boundary condition along $x$-direction.

where $F_{xy}$ is the lattice field strength defined as,

$$F_{xy}(p,q) = \ln U_x(p,q)U_y(p+1,q)U_x^{-1}(p,q+1)U_y^{-1}(p,q), \tag{17}$$

with $-\pi < \frac{1}{i}F_{xy}(p,q) \leq \pi$.

For the three gapped phases at densities 1/4, 1/2 and 3/4, we calculate the Chern number [26] for $W_0 > t$ to determine the topological nature of these phases. Despite of the presence of the edge states for the 1/4 (or 3/4) dimer insulator, the Chern number turns out to be zero for all of the three insulators.

As mentioned at the end of Section 4, the edge states are observed only along the zigzag edges of the lattice under open boundary condition along the $x$-direction. This is related to the fact that the dimer insulator structure can be effectively described as 1D SSH-like chains stacked in a 2D lattice, connected via weak hopping. Therefore, in order to probe the topological nature of the dimer insulators, we calculate Berry phase for each $k_y$ value separately according to the formula,

$$\gamma_q = \text{Im} \sum_{p=1}^{L_x} \ln U_x(p,q). \tag{18}$$

It turns out that the Berry phase ($\gamma$) is independent of $k_y$ (or $q$). Fig. 9 depicts the variation of the Berry phase as a function of $W_0/t$ calculated in the gapped region for three different densities $\rho = 1/4$, 1/2 and 3/4. We can see that for positive values of $W_0$ ($W_0 > t$), the Berry phase is quantized at $\pi$ for the dimer insulator at $\rho = 1/4$ and remains zero for the one at density 3/4. The situation is reversed when we reverse the sign of $W_0$ ($W_0 < -t$). On the other hand, in both of these cases $\gamma$ remains zero for the insulator at $\rho = 1/2$. This identifies the dimer insulators at $\rho = 1/4$ and 3/4 as weak topological insulators (WTIs) for $W_0 > t$ and $W_0 < -t$ respectively.

The existence of the WTI phase becomes clearer by realizing that the governing Hamiltonian obeys the following symmetries:

1. Time reversal symmetry: $TH(k_x, k_y)T^{-1} = H(-k_x, -k_y)$, where $T$ is the antiunitary time reversal operator $T = \mathcal{K}$, where $\mathcal{K}$ is the complex conjugation operator, satisfying $T^2 = 1$.

2. Mirror symmetry: $MH(k_x, k_y)M^{-1} = H(-k_x, k_y)$, with $M = \sigma_x \otimes \sigma_x$, using the notations of Appendix B.

The two symmetry operators $T$ and $M$ commute, $[T, M] = 0$, which places the model in the mirror symmetry class AI [27]. This symmetry class admits a $\mathbb{Z}$ topological number in 1D, but a zero topological number in 2D. Our model realizes a stacked WTI of such 1D chains.

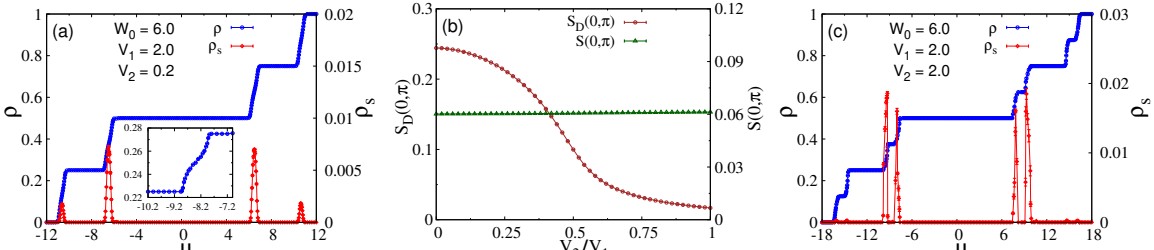

Figure 11: Variation of the density versus chemical potential in the presence of both NN repulsion $V_1$ and NNN repulsion $V_2$, with the on-site potential strength $W_0 = 6.0$, for (a) $V_1 = 2.0$, $V_2 = 0.2$, and (c) $V_1 = 2.0$, $V_2 = 2.0$. (b) Change of dimer structure factor and structure factor with the ratio $V_2/V_1$.

One way to interpret the presence of edge states for positive $W_0$ ($W_0 > t$) is illustrated in Fig. 3 for an effective model, which is described by underlying SSH chains for densities $\rho = 1/4$ (Fig. 3 a) and $\rho = 3/4$ (Fig. 3 c). Each chain admits alternating tunnelings $t$, $t_x$ along the chain, and neighboring chains are weakly-coupled via $t_y$. Depending on the sign of $W_0$, the chains for either $\rho = 1/4$ or $\rho = 3/4$ manifest their topological phase, which leads to edge states at the corresponding density, as long as $t_y$ is weak enough.

# 6 Effect of interactions

In this section, we discuss the effect of interactions between HCBs on the phase diagram and the WTI phase. First we consider NN repulsion between HCBs by adding

$$H_1 = V_1 \sum_{\langle i,j \rangle} \hat{n}_i \hat{n}_j, \tag{19}$$

to the Hamiltonian, Eq. (1). Fig. 10 compares the variations of the average density $\rho$ as a function of the chemical potential $\mu$ for three different values of $V_1$ with periodic and open boundary conditions.

The blue curves in Fig. 10 depict the average density of a $20 \times 20$ periodic honeycomb lattice, whereas the brown curves in the insets show the alteration of the plateau at $\rho = 1/4$ under open boundary condition. One can see that as we increase the NN repulsion the width of the plateaus corresponding to $\rho = 1/4, 1/2$ and $3/4$ increases. Since the band gap in any insulating phase is determined by the width of the plateau in the $\rho - \mu$ curve, the gaps corresponding to the three above-mentioned insulating phases simply gets larger for larger NN repulsion. In other words, the insulating phases become more stable in the presence of NN repulsion. This can be understood by realizing that the introduction of NN repulsion effectively increases the energy cost of adding another particle to the insulating structures. Therefore, energy minimization forces the system to be in the insulating phases for wider ranges of the chemical potential, thus increasing the band gap of these phases. Under open boundary condition along the $x$-direction, for each value of $V_1$, the plateau at $\rho = 1/4$ further splits into two plateaus corresponding to densities 0.225 and 0.275 similar to the case with zero NN repulsion. This means that the topological nature of the dimer insulator at $\rho = 1/4$ remains unaffected by the presence of NN repulsion.

We now argue that the WTI phase is in fact robust against any amount of NN repulsion, as elucidated by considering the spatial structure of the insulating phase. As discussed in Section 3, the WTI is a dimer insulator, where each dimer is formed by a particle hopping back and forth between the two sites belonging to a NN bond aligned along $x$-direction. Since

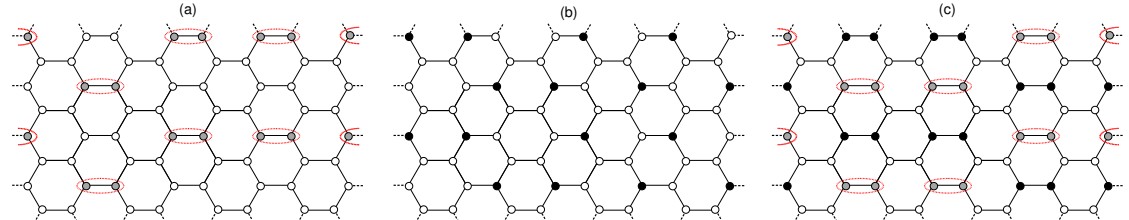

Figure 12: Pictorial description of the insulating structures corresponding to the density-plateaus in Fig. 11 c at (a) $\rho = 1/8$, (b) $\rho = 1/4$, and (c) $\rho = 3/8$.

the dimers are situated at alternate $y$-levels, the particles in two neighboring dimers do not feel any repulsion, as they never reside on two NN sites. Consequently, NN repulsion cannot restrict the hopping process involved in the formation of a dimer and thus the WTI remains uninfluenced.

This argument is also valid for the WTI phase at 3/4-filling, which is the particle-hole conjugate of Fig. 3 a. While NN repulsion is felt between the particle in each dimer and the particles frozen in the in-between layers (depicted by white circles in Fig. 3 a and reflecting filled sites in its particle-hole conjugate), this does not disrupt the formation of dimers. In fact this configuration is the minimum-energy configuration at density $\rho = 3/4$, even in presence of NN repulsion. While at this filling, the particle in each dimer encounters $2V_1$ repulsion from the two occupied NN sites, this would not depend on which of the two sites of the dimer it occupies. The particle will therefore prefer to hop back and forth between these sites, rather than choosing a particular site to reside in, thereby lowering the energy of the system.

The above discussion motivates the consideration of the effect of NNN interactions, described by an additional term

$$H_2 = V_2 \sum_{\langle\langle i,j\rangle\rangle} \hat{n}_i \hat{n}_j. \tag{20}$$

As depicted in Fig. 11 a, in presence of weak NNN repulsion the variations of the order parameters with the chemical potential remain almost unchanged for a periodic honeycomb lattice. With open boundary conditions the plateau corresponding to the dimer insulator at $\rho = 1/4$ splits into two parts (inset of Fig. 11 a) similarly to the non-interacting scenario. This demonstrates that the WTI phase is robust against small NNN repulsion values. Now, with the increase of the ratio $V_2/V_1$ the hopping process of the constituent particle of each dimer gets disrupted. Consequently, beyond some critical value of this ratio, it is energetically favorable for the particle to be localized at one of the two sites of the dimer instead of hopping back and forth. This way the particles can avoid NNN repulsion felt between two neighboring dimers along the $y$-direction. Such a configuration is depicted in Fig. 12 b. In Fig. 11 b the dimer structure factor $S_D(0, \pi)$ and the structure factor $S(0, \pi)$ are plotted as a function of $V_2/V_1$. We can see that with increasing value of $V_2/V_1$, the dimer structure factor decreases from 0.25 to a value close to zero, whereas the structure factor remains constant at 0.0625. Since the dimers are destroyed for larger values of $V_2$, the dimer structure factor obviously decreases. Nevertheless as the particle number in each $y$-layer is fixed the structure factor remains unaltered. Hence, it can be concluded that the WTI transforms into a normal insulator (similar to the one in Fig. 12 b) for larger values of NNN repulsion.

As can be seen from Fig. 11 c, the NNN interactions are observed to stabilize additional insulating plateaus at fillings 1/8, 3/8, 5/8 and 7/8 for a periodic honeycomb lattice. At $\rho = 1/8$ only half of the dimers in Fig. 3 a are formed so that no NN or NNN repulsion is felt between two HCBs. The structure corresponding to this insulator is not unique. One of its possible structures is demonstrated in Fig. 12 a for an $8 \times 8$ lattice. On the other hand,

at filling-fraction 3/8, half of the dimers of Fig. 3 a are occupied by an extra particle each, thus transforming a dimer into a pair of particles. Fig. 12 c depicts one of the many possible structures corresponding to this insulator. The dimers (pair of HCBs) in this insulating phase are distributed in a way such that no two dimers (pair of HCBs) are NNN of each other. Despite of the repulsion felt by the HCBs, this is in fact the minimum energy configuration of the system at $\rho = 3/8$. One should note that the insulator at density 5/8 (7/8) is exactly the same as the one at $\rho = 3/8$ (1/8) once the even and odd layers, as well as particles and holes, are interchanged. The detailed characterization of these additional insulating plateaus would be interesting to pursue in future.

## 7 Conclusions

To summarize, we have studied HCBs in a periodic honeycomb lattice with NN hopping ($t$) and alternating positive and negative on-site potential ($W_0$) along different $y$-layers, using SSE QMC supported by analytical calculations. The system reveals the existence of three insulating phases for $W_0 > t$: a CDW at 1/2-filling and two dimer insulators at densities 1/4 and 3/4. Depending on the on-site potential pattern, one of the dimer insulators turns out to be a WTI, with a zero Chern number but a non-trivial Berry phase, which is protected by mirror-symmetry and belongs to the mirror-symmetry class AI. The model can be effectively thought of as weakly coupled SSH chains, where the intermediate layers being either completely empty (for $\rho = 1/4$) or fully occupied (for $\rho = 3/4$). Although our study involves HCBs, it is important to note that the WTI phase persists in case of fermions as well. Since the energy bands are well-separated in the regime $W_0 > t$, the topological phase becomes oblivious to the exchange statistics of the constituent particles.

With recent advancements, optical lattice with ultra-cold atoms would be a perfect tool to actualize our model experimentally. Experiments on hexagonal lattices in this framework have already been around for a while [28–31]. The on-site potential of the lattice sites can also be tuned in these experiments, making it possible to achieve the pattern required by our model. Additionally, the measurements of Berry phase [32] as well as Chern number [33] in an optical lattice setup have also been performed successfully. Thus, we believe that our model is a promising and interesting candidate to realize weak topological insulating phase in optical lattice experiments. In addition, certain features of our model, including the band spectrum and the presence of edge states, could also be probed using driven-dissipative exciton-polariton microcavity lattices [34].

Besides the WTI phase, our model exhibits a rich phase diagram, which includes intriguing phases such as bosonic Dirac semi-metal and nodal line semi-metal among others. Since the main focus of our current work is the WTI phase, a detailed study of the other novel phases is outside the scope of this paper. It would be interesting to investigate these phases in more detail and examine how these phases are affected by the presence of off-site interaction in the system. Furthermore, it would be worthwhile to analyze the dependence of the phase diagram on the on-site repulsion $U$, when the HCBs are replaced by soft-core-bosons, as well as the persistence of the WTI in the $U \to 0$ limit.

## Acknowledgements

This research was funded by the Israel Innovation Authority under the Kamin program as part of the QuantERA project InterPol, and by the Israel Science Foundation under grant 1626/16. AG thanks the Kreitman School of Advanced Graduate Studies for support. AG also thanks M.

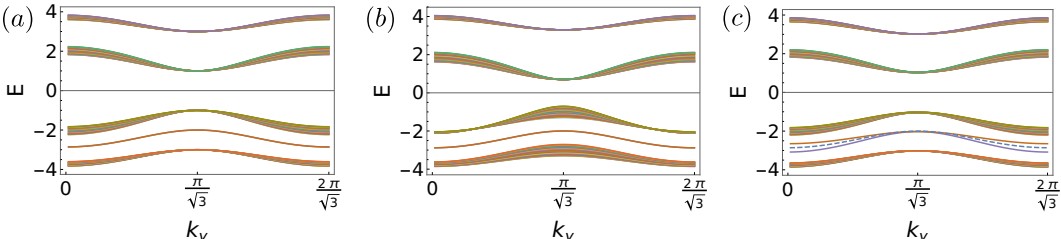

Figure 13: Protection of the edge states. In panel (a) we plot the spectrum of the model with open boundary conditions in the absence of perturbations. The edge states are visible near the center of the two lower bands. In Panel (b) and (c) we consider the effect of pertubations: (b) Mirror-symmetry preserving $g\gamma_{2,2}$, keeping the edge states intact; (c) Mirror-symmetry breaking perturbation $g\gamma_{3,1}$, for which the edge states split symmetrically around the unperturbed edge state (dashed line). Here $t = 1.0$, $g = 0.3$ and $W_0 = 2.0$.

Sarkar and S. Nag for useful discussions.

## A  The band structure

In this Appendix we present details of the band structure of the Hamiltonian in Eq. (1). The absence of interaction between two different HCBs makes it possible to express the Hamiltonian in single-particle basis. Consequently, the energy spectrum obtained from this Hamiltonian is the same for a HCB and a fermion. For a unit cell containing four sites, as depicted in Fig. 1, the Hamiltonian in Eq. (1) can be expressed in the momentum space as,

$$
H(\boldsymbol{k}) = \begin{pmatrix} -W_0 & t(1+e^{i\sqrt{3}k_y}) & 0 & te^{-i3k_x} \\ t(1+e^{-i\sqrt{3}k_y}) & W_0 & t & 0 \\ 0 & t & W_0 & t(1+e^{-i\sqrt{3}k_y}) \\ te^{i3k_x} & 0 & t(1+e^{i\sqrt{3}k_y}) & -W_0 \end{pmatrix}. \tag{21}
$$

The four branches of the energy spectrum corresponding to this Hamiltonian are,

$$
E_{\pm,\pm}(\boldsymbol{k}) = \pm[\epsilon(\boldsymbol{k}) \pm \eta(\boldsymbol{k})]^{1/2}, \tag{22}
$$

where

$$
\epsilon(\boldsymbol{k}) = W_0^2 + t^2 + 4t^2 \cos^2\left(\frac{\sqrt{3}}{2}k_y\right),
$$

$$
\eta(\boldsymbol{k}) = 2t\sqrt{W_0^2 + 4t^2\cos^2\left(\frac{3}{2}k_x\right)\cos^2\left(\frac{\sqrt{3}}{2}k_y\right)}. \tag{23}
$$

The Brillouin zone can be chosen to lie between $-\frac{\pi}{3} < k_x < \frac{\pi}{3}$ and $0 < k_y < \frac{2\pi}{\sqrt{3}}$. In order to study the energy spectrum, we divide the parameter space into three regions: $W_0 < t$, $W_0 = t$ and $W_0 > t$. For the half-filled system, we expect the lowest two bands $E_{-,+}$ and $E_{-,-}$ to be occupied. In the region $W_0 < t$, a pair of Dirac nodes appear in the energy bands $E_{-,-}$ and $E_{+,-}$ at $k_x = 0$ and two different $k_y$ values (Fig. 6 a). The positions of these nodes shift along the $k_y$-axis as a function of $W_0$ and $t$ values. This can be understood by realizing that to have

nodes in the $E_{\pm,-}$ energy bands at $k_x = 0$ we must have $\epsilon(0, k_y) = \eta(0, k_y)$, which can be simplified to the solution,

$$k_y^{\pm} = \pm\frac{1}{\sqrt{3}}\cos^{-1}\left[-\frac{W_0^2 + t^2}{2t^2}\right]. \tag{24}$$

Therefore, for $W_0 < t$ the half-filled system behaves as a bosonic Dirac semi-metal due to the presence of the pair of Dirac nodes at momenta $(0, k_y^{\pm})$.

Next, we note that at $k_y = \frac{\pi}{\sqrt{3}}$, for all possible values of $k_x$, we have

$$E_{-,-} = -\left(W_0^2 + t^2 - 2tW_0\right)^{1/2} = -|W_0 - t|, \tag{25}$$

and

$$E_{+,-} = +\left(W_0^2 + t^2 - 2tW_0\right)^{1/2} = |W_0 - t|. \tag{26}$$

At $W_0 = t$, the two point nodes thus coalesce and disappear at $k_x = 0$, $k_y = \frac{\pi}{\sqrt{3}}$ and a line node is formed at $k_y = \frac{\pi}{\sqrt{3}}$, as shown in Fig. 6 b. As a result, at half-filling we have a nodal line semi-metallic phase for $W_0 = t$.

To understand the situation for $W_0 > t$ of the half-filled system, it is important to note that, at $k_y = \frac{\pi}{\sqrt{3}}$ the lower band $E_{-,-}$ attains its maximum value $(-|W_0 - t|)$, while the upper band $E_{+,-}$ reaches its minimum $(|W_0 - t|)$. The band gap at half-filling is thus given by,

$$\Delta_{1/2} = E_{+,-} - E_{-,-} = 2|W_0 - t|. \tag{27}$$

Hence, for $W_0 > t$, the half-filled system develops a gap of width $2|W_0 - t|$ and becomes insulating (see Fig. 6 c).

The analysis of a 3/4-filled system for different values of $W_0$ and $t$ can be done by comparing the energy bands $E_{+,-}$ and $E_{+,+}$. The top of the lower band $E_{+,-}$ lies at $k_x = \frac{\pi}{3}$, $k_y = 0$, with the energy value

$$E_{+,-} = \left[W_0^2 + 5t^2 - 2tW_0\right]^{1/2}, \tag{28}$$

whereas the bottom of the upper band $E_{+,+}$ occurs at a different momentum $k_x = 0$, $k_y = \frac{\pi}{\sqrt{3}}$ with energy,

$$E_{+,+} = |W_0 + t|. \tag{29}$$

The indirect band gap is therefore expressed as,

$$\Delta_{3/4} = |W_0 + t| - \left[W_0^2 + 5t^2 - 2tW_0\right]^{1/2}. \tag{30}$$

So, the system at 3/4 filling-fraction will behave as an insulator as long as $\Delta_{3/4} > 0$, which simplifies to $W_0 > t$. For $W_0 = t$ we have $E_{+,-}(\frac{\pi}{3}, 0) = E_{+,+}(0, \frac{\pi}{\sqrt{3}}) = 2t$ and the band gap becomes zero, while for $W_0 < t$ the band gap $\Delta_{3/4}$ becomes negative. Consequently in the region $W_0 \leq t$, both of the energy bands $E_{+,-}$ and $E_{+,+}$ become partially filled and the system behaves as a bosonic semi-metal. A similar analysis of the system at 1/4 filling-fraction leads to the same conclusions as density 3/4.

## B   Protection of edge states

In this Appendix we study the formation of the edge states in the non-interacting version of our model. We take periodic boundary conditions along the $y$-direction and open boundary conditions along the $x$-direction. Fourier transforming along $y$ we obtain for each $k_y$ the Hamiltonian block

$$
\begin{aligned}
\hat{H}_{k_y}(x) = {} & -W(x)\gamma_{3,3} + t\gamma_{0,1}\big[1 + \cos(\sqrt{3}k_y)\big] \\
& - t\gamma_{3,2}\sin(\sqrt{3}k_y) + t\gamma_{1,1} + \frac{3t}{2}(\gamma_{1,2}+\gamma_{2,1})(-i\partial_x) \\
& - \frac{9t}{4}(\gamma_{1,1}-\gamma_{2,2})\partial_x^2,
\end{aligned} \tag{31}
$$

which is written in position represenation to second order in the momentum $\hat{p}_x$, in terms of the matrices $\gamma_{i,j} = \sigma_i \otimes \sigma_j$ (where $\sigma_i$ with $i = 1, 2, 3$ are the Pauli matrices and $\sigma_0$ is the $2\times 2$ identity matrix). This Hamiltonian admits a mirror symmetry $\mathcal{M}$ that is represented by $\mathcal{M} = U_{\mathcal{M}}\hat{\Pi}$ with $U_{\mathcal{M}} = \gamma_{1,1}$ and $\hat{\Pi}\hat{x}\hat{\Pi} = -\hat{x}$. We take $W(x) = -W_0\mathrm{sgn}(x - L/2)\mathrm{sgn}(x + L/2)$ which is chosen to respect the mirror symmetry while, for $W_0 > 0$, we get a topological phase for $|x| < L/2$ and a non-topological phase for $|x| > L/2$ at density $\rho = 1/4$ (and vice-versa for $\rho = 3/4$). Near $x = \pm L/2$ the step-like configuration localizes a single state $\psi_{\pm}(x)$ at the middle of the $\rho = 1/4$ gap at energy $-\varepsilon(k_y)$. Here $\varepsilon^2(k_y) \simeq W_0^2 + 2t^2[1 + \cos(\sqrt{3}k_y)]$ determines the dispersion of the edge state along the $y$-direction. The two edge states $\psi_{\pm}(x)$ satisfy $\mathcal{M}\psi_+(x) = \psi_-(x)$. They are distinct due to the localized nature of $\psi_{\pm}(x)$ (near $x = \pm L/2$), and occur at the same energy as $\mathcal{M}$ and $\hat{H}_{k_y}$ are commuting. Indeed, their degeneracy is maintained as long as no mirror symmetry breaking perturbations are added to the Hamiltonian, as demonstrated in Fig. 13 b.

While mirror symmetry is protecting the edge states from splitting, the edge states are further pinned to the center of the band due to an emergent chiral symmetry. This is revealed for the case of $\rho = 1/4$ by a projection to the lower two bands, which yields the effective Hamiltonian

$$
\hat{H}_{\mathrm{proj}} = -\varepsilon(k_y)\tau_0 + \varepsilon_x(k_x, k_y)\tau_x + \varepsilon_y(k_x, k_y)\tau_y. \tag{32}
$$

Here $\tau_0$ is the $2\times 2$ identity matrix and $\tau_i$ ($i = x, y, z$) are Pauli matrices in the basis of the two lower bands, $\varepsilon(k_y) \simeq W_0 + 2t_y(1 + \cos\sqrt{3}k_y)$ and

$$
\varepsilon_x(k_x, k_y) = t\cos(3k_x) + \frac{t_x}{2}\mathrm{Re}f(k_y), \tag{33}
$$

$$
\varepsilon_y(k_x, k_y) = t\sin(3k_x) + \frac{t_x}{2}\mathrm{Im}f(k_y), \tag{34}
$$

where $f(k_y) = \big[1 + \exp(-i\sqrt{3}k_y)\big]^2$, and $t_{x,y}$ are defined in Section 4. The Hamiltonian in Eq. (32) thus admits a partial chiral symmetry $\tau_z$, which manifests a particle-hole symmetry with respect to $-\varepsilon(k_y)$. This ensures that the average energy of the edge states stays pinned to the center of the band even in the presence of mirror symmetry breaking perturbations, see Fig. 13 c for an example of the resulting spectrum.

A similar argument holds for the case of $\rho = 3/4$ by projection to the upper two bands.

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
