# Peer review of "Weak topological insulating phases of hard-core-bosons on the honeycomb lattice"

_SciPost Physics, doi:SciPost Phys. 10, 059 (2021)_

## Round 2 · Referee Report · Anonymous (Referee 1) · 2020-11-30

Strengths

1 - The paper is very well written and easy to follow.
2 - The results are presented in an intuitive, pedagogical manner.
3 - The authors perform a detailed study of their model, using multiple order parameters, topological invariants, as well as varying boundary conditions.

Weaknesses

1 - The symmetry classification and topological protection of the model is insufficiently discussed (see report below).
2 - Is it not clear to what extent the work meets the acceptance criteria of Scipost Physics (specifically, the list of "Expectations"), as opposed to Scipost Physics Core.1 - The symmetry classification and topological protection of the model is insufficiently discussed (see report below).
2 - Is it not clear to what extent the work meets the acceptance criteria of Scipost Physics (specifically, the list of "Expectations"), as opposed to Scipost Physics Core.

Report

The authors study the topological phases of hard-core bosons on a hexagonal lattice in which the onsite potential is modulated. They find that WTI phases appear once the onsite potential is larger than the nearest neighbor hopping strength, and that these phases are robust against NN repulsion as well as against weak NNN repulsion.

The paper is very well written. I enjoyed reading it. Results are presented in an intuitive, pedagogical way, making them easy to follow. There are however two points that I think the authors should address. These points are listed above, and I detail them here:

1) The authors discuss WTI phases appearing in symmetry class BDI and use a Hamiltonian that is non-interacting (I'm referring to Eq. 1, before the NN and NNN repulsion are added). However, the single-particle Hamiltonian of Eq. 1 does not belong to symmetry class BDI. It does have time-reversal symmetry T=K, meaning it is real, but there is no chiral symmetry. There is no unitary that anti-commutes with H because of the non-zero onsite modulation and chemical potential. Consistent with this lack of chiral symmetry, the edge states discussed by the authors do not appear at E=0. In class BDI, it is not just translation symmetry, but also chiral symmetry which protects the WTI. Because of chiral symmetry all states come in +E, -E pairs (as can be seen from the bandstructures of Fig. 6). The edge states of a WTI should be pinned to the middle of the E=0 gap, such that they cannot be removed from this gap without breaking symmetries. In the authors' model however, edge states appear in the gap between bands 1 and 2 (or 3 and 4), away from E=0. What symmetry is responsible for their topological protection? Why can't they, in principle, be shifted up or down in energy such that they hybridize with the bulk states and dissapear?

2) While the work is novel and well presented, the authors should spend more time discussing if/how their paper meets the expectations of Scipost Physics (https://scipost.org/SciPostPhys/about#criteria). From my reading of the paper as it is now, it seems to me that it instead meets the acceptance criteria of Scipost Physics Core (https://scipost.org/SciPostPhysCore/about), provided that the point (1) above is addressed.

Requested changes

1 - Show explicitly what are the symmetries of their model and its symmetry class.
2 - Prove that their phases are topologically protected. This means to prove that there does not exist a symmetry-preserving perturbation which removes the edge states, for instance by shifting their energies away from their respective gaps.

---

## Round 2 · Referee Report · Anonymous (Referee 2) · 2020-12-9

Report

This manuscript studies a bosonic analogue of weak topological insulating phases on the honeycomb lattice. Specifically the model studied are on a two-dimensional periodic honeycomb lattice in the presence of an on-site potential with alternating sign along the different y-direction of the lattice. Using quantum Monte Carlo simulations and analytical calculations, the authors identify a bosonic weak topological insulator, characterized by a zero Chern number but non-zero Berry phase, which is manifested at either density 1/4 or 3/4, as determined by the potential pattern. They also map out the full phase diagram, including a charge-density-wave insulator at 1/2-filling and superfluid at intermediate densities. Supprisely the weak topological insulator is further shown to be robust against any amount of nearest-neighbor repulsion, as well as weak next-nearest-neighbor repulsion. The proposed model may be experimentally realized using cold atoms in an optical lattice. I find the results are interesting from both theoretical and experimental aspects, so I recommend its publication.

I have the following comments:

1: a main character of weak topological insualtor is the existence of edge states on the edges along specific directions. Here the edge state is quasi-1D superfluid. One may calculate the single-particle correlator b^{dagger}_i b_j. The decaying behavior may reflect such information: it is insulating if the decay is exponential with the distance, and is gapless superfluid if the decay follows a power law.

2: Since the authors are studying a bosonic model, the Chern number and Berry phase for bosons should be calculated to characterize the bosonic weak topological insulator. as well as weak next-nearest-neighbor repulsion. The experimental realization of our model is feasible in an optical lattice setup.

---

## Round 3 · Referee Report · Anonymous (Referee 1) · 2021-2-10

Report

The authors have fully addressed the points I raised in the first round. I believe this paper should be published in SciPost Physics as is.

---

## Round 3 · Author Response

Dear Editor,

We thank the Referees for their positive evaluation and constructive comments, and for recommending publication.

We have modified the paper according to the Referees’ comments, which we believe has improved the paper considerably. In particular, we have modified the introduction to stress the context of the results. We believe that our work offers a natural, almost minimal model for the realization of weak topological insulators (WTIs) of interacting bosons in 2D. Indeed, a study of such bosonic WTIs and their associated phase diagrams has been lacking in the literature and we believe that our model fills this important gap by studying the nucleation of these topological phases, their stability properties in the presence of interactions, and the interplay of the various competing orders. In addition, our paper opens up new pathways to the study of bosonic topological states by identifying the relevant tools and order parameters and by generating a framework for their study. (Indeed we already followed up on this paper with an additional paper and we see potential for an extensive theoretical study of weak and strong topological states of bosons.)

We hope that the paper is now ready for publication. Below we enclose a point by point response to the Referees.

Sincerely, Amrita Ghosh and Eytan Grosfeld

Response to Referee 1

  • "Show explicitly what are the symmetries of their model and its symmetry class."

We thank the Referee for this comment. The only symmetries in the model are time-reversal symmetry and mirror symmetry. The model is therefore classified in the AI mirror-symmetry protected class, which admits a topological number in 1D. We now detail the symmetry classification towards the end of section V.

  • "Prove that their phases are topologically protected. This means to prove that there does not exist a symmetry-preserving perturbation which removes the edge states, for instance by shifting their energies away from their respective gaps."

We added an appendix B that describes the edge-bulk correspondence and explains the protection of the edge states by a mirror symmetry and an emergent chiral symmetry. We added a figure that demonstrates the robustness of the edge states in the presence of mirror-symmetry preserving perturbations and their splitting when mirror symmetry is broken. Due to the emergent chiral symmetry, the average energy of the edge states stays pinned to the center of the rho=1/4 (or rho=3/4) gap.

Response to Referee 2

  • "A main character of weak topological insualtor is the existence of edge states on the edges along specific directions. Here the edge state is quasi-1D superfluid. One may calculate the single-particle correlator b^{dagger}_i b_j. The decaying behavior may reflect such information: it is insulating if the decay is exponential with the distance, and is gapless superfluid if the decay follows a power law."

We thank the referee for this comment. Calculating directly the correlation function along the edge is difficult using SSE QMC. Instead, in order to verify the superfluid nature of the edge state, we calculated the superfluid density along the y-direction for the different stripes of the new Figure 1b. We observe that the superfluid density is finite along the edges of the sample but is vanishingly small (zero in the thermodynamic limit) in the bulk. This is summarized in a new figure 7c and referred to in the text.

  • "Since the authors are studying a bosonic model, the Chern number and Berry phase for bosons should be calculated to characterize the bosonic weak topological insulator. "

We have calculated the Chern number and Berry phase for both bosons and fermions and found that they give the same value for both types of particles, see Eqns. (14,15) and the nearby discussion. We added a sentence to the introduction to stress this.

---

## Round 3 · List of Changes

1. Introduction modified to stress the context of the results and the importance of our study.
  2. In a new Figure 1 we now label the stripes along the y-direction.
  3. In a new Figure 7c we now plot the superfluid density along the y-direction for the different stripes, for the topological and non-topological phases. We discuss this in the text in Section 4.
  4. In Section 5 we discuss the symmetries of the model.
  5. In a new Appendix B we demonstrate the protection of the edge states via mirror and effective chiral symmetries. A new Fig. 13 demonstrates this protection in the presence of several types of perturbations.
  6. We added two new references related to the symmetry class and to a new study of bosonic topological phases by us.

---

## Editorial Decision

published